# An Environmental and Economic Analysis of Strawberry Production in Southern Italy

Maria Pergola [1], Angela Maffia [2,*], Giuseppe Carlucci [3], Alessandro Persiani [4], Assunta Maria Palese [4], Massimo Zaccardelli [5], Gessica Altieri [6] and Giuseppe Celano [1]

1 Degree Course of Agriculture, Dipartimento di Farmacia, Università degli Studi di Salerno, Via Giovanni Paolo II, 132, 84084 Fisciano, Italy; mpergola@unisa.it (M.P.); gcelano@unisa.it (G.C.)
2 PhD School, Mediterranea University, 89124 Reggio Calabria, Italy
3 Agreenment s.r.l., 75100 Matera, Italy; gius.carl80@gmail.com
4 Ages s.r.l. s., 85100 Potenza, Italy; persianialessandro@gmail.com (A.P.); palesedina@gmail.com (A.M.P.)
5 Centro di Ricerca Orticoltura e Florovivaismo (CREA), Via Cavalleggeri, 25, 84098 Pontecagnano, Italy; massimo.zaccardelli@crea.gov.it
6 PhD School of Agriculture, Forestry, Food and Environmental Sciences, Università degli Studi della Basilicata, Via Nazario Sauro 85, 85100 Potenza, Italy; gessica.altieri@unibas.it
* Correspondence: angela.maffia@unirc.it; Tel.: +39-380-741-3533

**Abstract:** This paper aims to provide an evaluation of the environmental and economic aspects of strawberry cultivation in the Campania and Basilicata regions of Southern Italy, and to consider the effects on strawberry productivity following compost tea (CT) application. Eight strawberry-growing systems were tested. To this end, compost tea production and characterization were described; a quantitative analysis of the strawberries' yield was performed, and environmental impact per ha and per kg of strawberries was estimated using the life cycle assessment methodology. To compare the profitability of the systems analyzed, the gross profit of the farmers was calculated, also considering the social cost of pollution. One of the two organic systems analyzed, using solarization for soil disinfestation, biological fight for pest control, and corrugated boxes as packaging recycled at the end-of-life, was the most sustainable system with carbon credits. At the same time, organic crops are not always the most sustainable and profitable systems if significant irrigation and fertigation interventions are carried out, as in another organic system analyzed. Plastic materials and zinc structures were the most impacting items in almost all analyzed systems. The use of a CT with an elevated number of beneficial microorganisms with a high suppressive action allowed to obtain a good increase of the yield, in both systems that used it, and to have a higher gross profit. On the other hand, the validity of this technique was strongly linked to the finding of high-quality green compost.

**Keywords:** sustainability; life cycle assessment; environmental prices; economic analysis; pollution cost; circular economy; compost

## 1. Introduction

In Italy, strawberry production occurs above all under tunnels (greenhouses), on about 268,000 hectares [1] for a total value of € 360 million. Campania and Basilicata are the regions with the largest cultivated areas (103,000 and 38,600 ha, respectively) and the highest strawberry productions (43,700 and 12,100 t, respectively) [1].

In 2022, the expenditure of Italians for the purchase of strawberries increased both compared to 2021 (+4.1%) and 2019 (+23%). These increases were essentially determined by the increase in the average price which grew in 2022 by 4.6% compared to 2021 and by 20% compared to 2019 [2].

In recent years, more and more producers and consumers are realizing the environmental impact of goods production processes. This awareness leads them to look for more sustainable solutions [3] and home behaviors that top the list of the most common climate

actions, which are recycling/composting, saving energy and water, and avoiding food waste. From a survey launched by the World Economic Forum and Ipsos in 2021, it was found that on average, more than half of the interviewed have modified their consumer behavior out of concern about climate change over the past few years, and women seem to be more likely than men to change their consumption patterns [4].

In agriculture, composting is one of the most reliable, common, simple, and sustainable solutions to managing agricultural and agribusiness waste, easily achievable on farms [5]. This practice allows, under controlled conditions, to quickly transform organic waste into a final relatively stable organic product—named compost—free from animal and plant pathogens and like soil humus. Therefore, compost provides organic matter and nutrients to the amended soils. According to the circular economy perspective, composting favors the recycling of waste which becomes an important production factor within the agricultural farm [6]. Starting from a good compost, compost tea (CT) is produced and defined as a "liquid organic-filtered formulates produced by immersion, extraction, and oxygenation of a compost in a liquid, generally water, for a period ranging from few hours to two weeks, with or without additives, and in absence of any solvent" [7,8]. Several studies have shown that compost applications can improve growth and quality, also in strawberry production. Indeed, Hargreaves et al. [9] analyzed the efficacy of compost tea infusions made with municipal solid waste compost (MSWC) and ruminant compost as amendments in strawberry cultivation. Arancon et al. [10] evaluated the effects of the application of vermicompost processed commercially from food and paper wastes on the growth and yields of strawberries. Duffy et al. [11] monitored the potential risk for regrowth and transmission to strawberry plants of *E. coli* O157:H7 and Salmonella Thompson in teas made from various types of compost. Hargreaves and Warman [12] studied for two years the influence of the application of three levels of MSWC and two application rates of CT made with MSWC to strawberries. Welke [13], in an organic market garden farm in the Southern interior of British Columbia, investigated the effect of compost extracts on strawberry yields and in the suppression of grey mold, *Botrytis cinerea*, over a period of two growing seasons.

At the same time, as for other agri-food productions, the assessment of the environmental impacts of strawberry production is important to identify more sustainable production practices and to guide consumers in their choices [3]. One of the most widely used methodologies to estimate environmental impacts is life cycle assessment (LCA), a "cradle-to-grave methodology to assess products, processes, services, activities, and systems based on the life cycle thinking approach" [14]. LCA has been proven as a valuable tool to address the questions about the environmental impact of different agricultural production systems [15], relying both on the identification of the subsystems that contribute most to the total environmental impact and the comparison of products and processes with the same functions [16–22]. In the literature there are various LCA studies for strawberry production: in Germany, Galafton et al. [23] estimated the environmental impacts of various plasticulture methods to help farmers determine the most environmentally friendly strawberry cultivation technique and to test the inclusion of plastic pollution in LCA; in Kentucky (USA), Clark and Mousavi-Avval [24] evaluated the global warming potential (GWP) of organic strawberries grown under high tunnels using the LCA methodology; in California, Parajuli et al. [25] performed a cradle-to-grave LCA study for an open-field strawberry including the impacts of food waste generated at each step of the life cycle; in Tunisia, Bakari et al. [26] assessed the impact of irrigation with treated wastewater at different dilutions on growth, quality parameters and contaminants transferred in strawberry fruits and soil; Ilari et al. [27], in central Italy, compared the environmental impact of two strawberry cultivation systems (a mulched soil tunnel and a soilless tunnel system); Legua et al. [28] applied the LCA to identify the potential environmental impact of dredged sediments used as growing media for strawberries; Romero-Gámez and Suárez-Rey [29] evaluated the environmental footprint of different strawberry production systems in Spain; Valiante et al. [30] evaluated the environmental impact of strawberry production in Italy

and Switzerland using the LCA approach; in Italy, Girgenti et al. [31] indicated that the majority of the greenhouse gas (GHG) emissions could be attributed to the plastic used in the production phases; in Iran, Khoshnevisan et al. [32] evaluated the environmental impacts of open field and greenhouse strawberry production; similar studies were performed to assess impacts of strawberry production in Australia [33], the United Kingdom and Spain [34]. However, to our knowledge, there is no LCA study on strawberries evaluating together the environmental impact and the profitability of strawberry production considering the costs that society must bear for pollution.

In light of what has been said so far, the present study aims to provide an evaluation of the environmental and economic aspects of strawberry cultivation in two regions of Southern Italy (Basilicata and Campania), considering the social cost of pollution, and to assess the effects on strawberry productivity following the application of the compost tea in two fields, one characterized by replanted rows and another by not replanted rows.

## 2. Materials and Methods

### 2.1. Systems Description

This study was carried out in Southern Italy specifically in Naples, Salerno, Caserta Provinces (Campania region), and in Matera Province (Basilicata region). Strawberry systems were cultivated in double rows on each bed.

Experimental strawberry systems in the Campania region were as follows:

- Conventional (SC), growing on rows made ex novo and managed during the production cycle according to the ordinary cultivation techniques;
- Integrated (SI), growing on rows made ex novo and managed during the production cycle according to the integrated cultivation techniques [35];
- Organics (SO1, SO2), growing on rows made ex novo and managed during the production cycle according to the organic cultivation techniques [36].

Strawberry systems studied in the Basilicata region were:

- Replanted Strawberry Crop treated with Compost Tea (RSC + CT)—strawberry plants were grown on rows already used in the previous production cycle. Ordinary cultivation techniques were carried out supplemented by seven root applications (through fertigation) of CT produced on farms;
- Replanted Strawberry Crop without Compost Tea application (RSC)—strawberry plants were grown on rows already used in the previous production cycle. Ordinary cultivation techniques were performed;
- Not Replanted Strawberry Crop treated with Compost Tea (NRSC + CT)—strawberry plants were grown on rows made ex novo. Furthermore, in this case, the ordinary cultivation techniques were implemented during the production cycle and supplemented by seven root applications (via fertigation) of CT produced on farms;
- Not Replanted Strawberry Crop without Compost Tea application (NRSC)—Strawberry plants were grown on ex novo rows and managed during the production cycle according to ordinary cultivation techniques.

The main characteristics of the studied strawberry systems are reported in Table 1. These data were acquired in the last two cropping years by direct interviews with the five farmers where the analyzed systems were located, and part of two specific research projects involving them and the authors of the paper; consultation of their field notebooks and visits to the farms. A specific collection sheet was prepared in order to acquire the information necessary for this study.

**Table 1.** Technical and agronomic characteristics of the analyzed strawberry systems (SC: conventional system; SI: integrated system; SO1: organic system; SO2: organic system; RSC + CT: replanted strawberry crop treated with compost tea; RSC: replanted strawberry crop without compost tea application; NRSC + CT: not replanted strawberry crop treated with compost tea; NRSC: not replanted strawberry crop without compost tea application).

| Orchard Characteristics | SC | SI | SO1 | SO2 | RSC + CT | RSC | NRSC + CT | NRSC |
|---|---|---|---|---|---|---|---|---|
| Cultivar | *Melissa, Sabrina, Flaminia, Nabila* | *Melissa, Sabrina* | *Melissa* | *Sabrina, Marinbella, Savana* | *Sabrosa, Rossetta* | | | |
| Planting density (plants ha$^{-1}$) | 75,000 (0.35 m × 0.20 m) | 99,500 (0.30 m × 0.30 m) | 99,500 (0.30 m × 0.30 m) | 70,000 (0.20 m × 0.30 m) | 75,000 (0.20 m × 0.30 m) | | | |
| Soil texture | | Sandy—silty | | | Sandy | | | |
| Cultivation system | Conventional under greenhouse | Integrated under greenhouse | Organic under greenhouse | Organic under greenhouse | Integrated under greenhouse | | | |
| Irrigation | Drip line | Localized | | | Drip line | | | |
| Fertilization | Mineral | Mineral/Organic | Mineral/organic/ green manure | Green manure/Organic | Mineral + Organic | Mineral | Mineral + Organic | Mineral |
| Soil management/Weed control | Not applied | Not applied | Not applied | Manual | Glyphosate | | | |
| Soil disinfestation/Disease control | Conventional products | Organic products | Organic products | Solarization/ Biological fight | Conventional products | | | |
| Harvesting method | | | | Manual | | | | |

## 2.2. Compost Tea Production and Characterization

CT was produced by an aerobic fermentation through an extractor in a liquid phase made with farm equipment easy to find, to ensure simplicity of management and high technological transferability. The compost utilized was a green composted soil improver, coming from the composting of vegetable waste (in particular, crop residues from fruit and vegetable processing plants), whose specifications can be found in Pergola et al. [5,37,38]. Analytical characteristics [39] of the green compost used, referring to the product screened at 2 cm, are shown in Table 2.

**Table 2.** Analytical characteristics of the compost (Dry Weight—DW) used to produce compost tea.

| Parameters | Values (% DW) | Parameters | Values (mg kg DW$^{-1}$) |
|---|---|---|---|
| Ashes | 61.80 | N-NH$_4$ | 167.00 |
| CaCO$_3$ | 7.63 | N-NO$_3$ | 478.00 |
| C | 20.56 | Ni | 37.31 |
| Total N | 1.55 | Cr | 66.28 |
| H | 3.27 | Zn | 140.72 |
| HA-C | 5.29 | B | 85.44 |
| FA-C | 1.31 | Mn | 705.78 |
| Humic-C | 6.60 | Cu | 114.86 |
| P | 0.62 | **Parameters** | **Values (mS cm$^{-1}$)** |
| C/P | 33.20 | Conductivity | 3.95 |
| S | 0.61 | **Parameters** | **Values** |
| Ca | 5.11 | pH | 10.70 |
| K | 3.18 | C/N | 13.30 |
| Na | 1.73 | | |
| Mg | 1.17 | | |
| Fe | 2.07 | | |

The CT production process involved the use of a self-made bio-extractor (a low-cost artifact) consisting of a sturdy cubic tank in polyethylene with a capacity of 1000 L, already available on the farm, capable of containing the liquid mass. The tumultuous oxygenation and mixing system was created through the installation of an electric compressor. While fine oxygenation was ensured by a small air generator (a very simple air pump generally used in aquariums) with attached rubber pipes, solenoid valves, and timers were used to time the active injection of compressed air on a periodic basis. After assembling the bio-extractor, jute bags containing the green compost to be extracted were placed in the tank filled with water. The compost/extractant ratio was equal to 1:8 *v/v* (Figure 1).

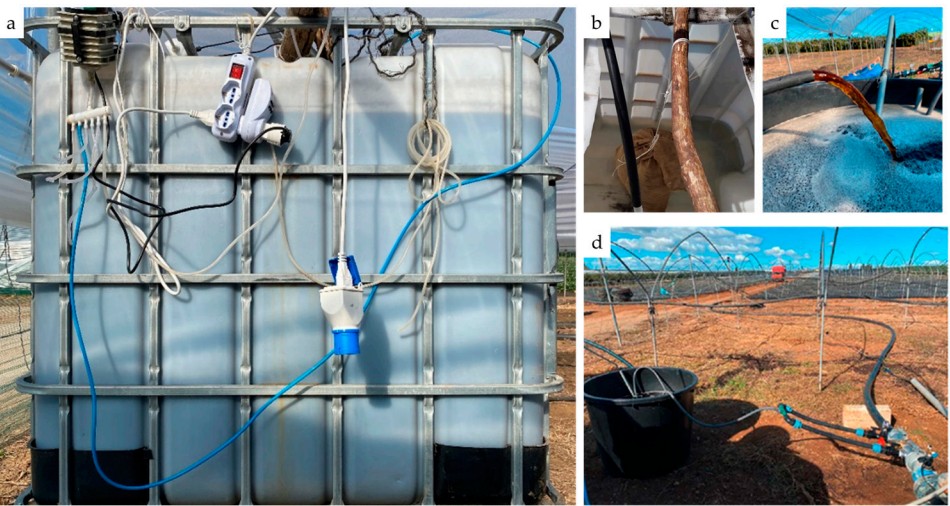

**Figure 1.** (**a**) The bio-extractor used to produce the compost tea; (**b**) the packaging of the jute bags containing the compost and the positioning inside the tank with consecutive addition of water; (**c**) compost tea emptying operation; (**d**) distribution of the compost tea through fertigation devices/systems.

The CT extraction took five days. During the extraction phase, compressed air was blown into the aqueous suspension through the air pump for 15 min every 3 h. A more energetic ventilation was made for 1 min every 6 h through the electric compressor. At the end of the extraction process, the aqueous suspension was filtered by a nylon filter and poured into a container. Then the ready CT was distributed to the plants through the already existing irrigation devices/systems. The distribution of CT took place through 7 fertigations, which were repeated every 15 days. CT was diluted in water with a ratio of 7:100 *v/v* just before application to the plants (Figure 1).

Before being used in the cultivation of strawberries, CT was subjected to laboratory analysis for the microbiological and chemical characterization (*Escherichia coli*; pH; EC—$\mu$s cm$^{-1}$; Yeasts; Molds; *Bacillus*; *Pseudomonas*; Total bacteria) [6,40–42].

Assays on seeds of *Eruca sativa* were carried out to determine the phytotoxicity/biostimulation action effects of the CT sampled at the end of the extraction. Five replicates were tested. Ten seeds were placed in 10 cm Petri dishes, containing 10 mL of CT and a paper filter. The control was performed in five replicates, using ultrapure water. The seeds were incubated for 72 h in a dark environment at 25 °C. At the end of the test, the germinated seeds were counted and epigeal part length (A, mm) and root extension (R, mm) were measured using standard procedures [39].

### 2.3. Environmental Analysis

According to the ISO 14040 [43] and the ISO 14044 [44], the LCA methodology was used to assess the environmental impacts of the analyzed systems. Following its four interrelated phases (1—goal and scope definition; 2—life cycle inventory; 3—life cycle impact assessment; 4—interpretation) the aims of this analysis were the evaluation of the envi-

ronmental sustainability of the strawberries production in two different regions and the sustainability assessment of the use of CT. The reference period of the analysis was set to the end of two production cycles to better test the effects of the use of compost tea on the crop, particularly when this occurs on rows already used in the previous production cycle (RSC + CT). The system boundaries went from the extraction of raw materials (inputs) to the farm gate (fruit harvesting), excluding the crop rotation used to improve soil fertility, because this study intended to compare the different stages of the agricultural process to identify the weakest link in the productive chain. All inputs (fuel, lubricants, fertilizers, pest control products, water, materials for setting up the irrigation system, etc.) were included considering their manufacturing processes. As functional units (FU), 1 kg of harvested fruits and 1 hectare of farmland were chosen as in other studies [45–48].

Material input types and the amounts used (primary data) were given priority as in previous studies [45,46]. Thus, Table 3 reports the life cycle inventory data of the investigated systems: amounts of fertilizers, chemicals, diesel fuel, water, and other items. Such information was acquired in situ during the last two agricultural years using a data collection sheet. The following farming operations were considered: plantation (which includes soil disinfection and preparation, pre-plant fertilization, tree plantation, etc.); soil tillage; fertilization; disease control; irrigation; other crop-specific operations, and harvesting. For each operation, to estimate direct and indirect emissions, the active ingredient of each product, as well as the amounts of the consumed fuel, water, and energy, were taken into account for calculation and used in the analysis.

**Table 3.** Farm inputs used in the analyzed strawberry systems. Two years average values (SC: conventional system; SI: integrated system; SO1: organic system; SO2: organic system; RSC + CT: replanted strawberry crop treated with compost tea; RSC: replanted strawberry crop without compost tea application; NRSC + CT: not replanted strawberry crop treated with compost tea; NRSC: not replanted strawberry crop without compost tea application).

| | SC | SI | SO1 | SO2 | RSC + CT | RSC | NRSC + CT | NRSC |
|---|---|---|---|---|---|---|---|---|
| **Fertilizers (kg ha$^{-1}$)** | | | | | | | | |
| Agristart magnum | | | | | 76 | 76 | 76 | 76 |
| Simple phosphate mineral fertilizer | | | | | 55 | 55 | 55 | 55 |
| Simple mineral nitrogen fertilizer | | | | | 83 | 83 | 83 | 83 |
| Simple mineral potassium fertilizer | | | | | 107 | 107 | 107 | 107 |
| Vegetable/organic fertilizer | 1500 | | | | | | | |
| Calcium nitrate | 30 | 550 | | | | | | |
| Potassium nitrate | 25 | | 350 | | | | | |
| Ammonium nitrate | 300 | 50 | | | | | | |
| Magnesium nitrate | | 100 | | | | | | |
| Urea | 75 | 450 | | | | | | |
| Magnesium sulfate | | 250 | 2300 | | | | | |
| Potassium sulfate | | | 2326 | | | | | |
| Ammonium sulfate | | | 200 | | | | | |
| Hydromix | | 25 | 1290 | | | | | |
| Resolvine 500 | | 15 | 65 | | | | | |
| Siapton | | 40 | 6265 | | | | | |
| Fertildung stallatico | | | 20 | | | | | |
| Auxine e citochinine | | | 1584 | | | | | |
| Solubordf | | 11 | | | | | | |
| Partner 700 | | 30 | | | | | | |
| Nutrimix | | 10 | | | | | | |
| Novatecsolub | | 150 | | | | | | |
| Maxiron | | 8 | | | | | | |
| Fosfoman | | 120 | | | | | | |
| Calcio bio | | 10 | | | | | | |
| Biocal | | 4 | | | | | | |
| Bioup | | 0.5 | | | | | | |

**Table 3.** *Cont.*

| | SC | SI | SO1 | SO2 | RSC + CT | RSC | NRSC + CT | NRSC |
|---|---|---|---|---|---|---|---|---|
| Chelated iron | | | | 1 | | | | |
| N | | 85 | 85 | | | | | |
| P$_2$O$_5$ | | 45 | 45 | | | | | |
| K$_2$O | | 90 | 90 | | | | | |
| Mg | | 8 | 8 | | | | | |
| **(L ha$^{-1}$)** | | | | | | | | |
| Lisofert biogarder | | | | 42 | | | | |
| **(m$^3$ ha$^{-1}$)** | | | | | | | | |
| Compost tea | | | | | 15.68 | | 15.68 | |
| **Chemicals (kg ha$^{-1}$)** | | | | | | | | |
| Affirm | | | | | 1.5 | 1.5 | 1.5 | 1.5 |
| Laser | | | | | 0.25 | 0.25 | 0.25 | 0.25 |
| Ortiva | | | | | 1 | 1 | 1 | 1 |
| Topas | | | | | 1 | 1 | 1 | 1 |
| Dargonis | | | | | 0.6 | 0.6 | 0.6 | 0.6 |
| Nimrod | | | | | 1 | 1 | 1 | 1 |
| Signum | 3 | | | | | | | |
| Copper oxychloride | | 4.2 | 4.2 | | | | | |
| Metaldeide | | | 5 | | | | | |
| Sulfur | | 4 | 4 | | | | | |
| **(L ha$^{-1}$)** | | | | | | | | |
| Epik | | | | | 2 | 2 | 2 | 2 |
| Roundup | | | | | 5 | 5 | 5 | 5 |
| Chloropicrin | 180 | | | | | | | |
| Dichloropropene | 240 | | | | | | | |
| Piretrine | | 4.6 | 4.6 | | | | | |
| Spinosad | | 0.4 | 0.4 | | | | | |
| Piraclostobin | | 3 | 3 | | | | | |
| Clofentezine | | | 0.3 | | | | | |
| Pirimetalin | | | 2 | | | | | |
| Micronized sulfur | | | 1.2 | | | | | |
| **Iron structures (kg ha$^{-1}$year$^{-1}$)** | 678 | 807 | 807 | 870 | 610 | 610 | 1219 | 1219 |
| **Plastics (films, pipes, containers) (kg ha$^{-1}$year$^{-1}$)** | 1900 | 3402 | 2578 | 1352 | 2150 | 2150 | 2300 | 2300 |
| **Human labor (h ha$^{-1}$)** | 4063 | 6150 | 3192 | 788 | 266 | 259 | 287 | 280 |
| **Machinery (h ha$^{-1}$)** | 52 | 51 | 35 | 22 | 74 | 74 | 100 | 100 |
| **Diesel (kg ha$^{-1}$)** | 234 | 201 | 201 | 179 | 46 | 46 | 69 | 69 |
| **Water (mc ha$^{-1}$)** | 1440 | 3000 | 3000 | 1133 | 9000 | 9000 | 9000 | 9000 |
| **OUTPUT—Strawberries average yield (kg ha$^{-1}$)** | 38,000 | 43,550 | 24,000 | 52,500 | 47,850 | 41,250 | 52,200 | 45,000 |

The estimation of direct emissions was performed using SimaPro's LCI databases, regarding fuel and lubricants; the methodology of Brentrup et al. [49] and that of IPCC [50], regarding fertilizers; and the method suggested by Hauschild [51] for emissions by synthetic pesticides released into the air, surface water, groundwater [22].

The embodied emissions, related to the production of electricity, diesel, lubricants, fertilizers, pesticides, and the construction of plastic, cardboard, wooden containers, and fixed structures (irrigation systems and supporting and covering structures), were assessed using the international database, Ecoinvent 3 [52]. A detailed description of the procedure is reported in the study carried out by Pergola et al. [46].

The impact assessment was performed with the SimaPro 9 software, according to the Environmental prices method developed by CE Delft [53], which expresses environmental impacts in monetary units. The characterization step was based on ReCiPe's (2008) Midpoint, hierarchic perspective [54], except for climate change, which was on the contrary based on IPCC 2013 methodology, as prescribed by the developers. The environmental

prices were not available for the following impact categories: natural land transformation, water, metal, and fossil depletion. Consequently, this method first assesses the environmental impacts and then expresses the social cost or pollution in Euros per kilogram pollutant indicating the loss of economic welfare that occurs when one additional kilogram of the pollutant finds its way into the environment [55].

*2.4. Economic Analysis*

The objective of the economic analysis was to compare the profitability of the eight analyzed systems. For this purpose, technical coefficients (yields, labor, and material inputs, use of fixed capital) were collected from the experimental fields throughout the 2020–2022 period and then they were converted into economic information, imputing prices and tariffs recorded on the marketplace of Matera, Naples, Salerno and Caserta in the 2022/2023 harvesting campaign [55]. Processed data were integrated with additional information acquired after sample surveys on local farms and then validated through interviews with agricultural stakeholders, extension services, and market operators.

Economic results were expressed at constant values and the profitability of the strawberry systems analyzed was compared through the gross profit (GP) of the farmer, obtained by deducting the revenue from sales of strawberries (Total Output—TO) from all variable production costs (VC), gross of taxes and overheads. A detailed description of the economic analysis carried out can be found in Palese et al. [56].

*2.5. Statistical Analysis*

The significant differences ($p < 0.01$) between the parameters under study were evaluated using the control-Dunnett test.

**3. Results**

*3.1. Characterization and Biostimulative Action of the Compost Tea Used*

The analyses performed on the CT showed the absence of pathogens such as *Escherichia coli* and a high number of beneficial microorganisms (*Bacillus, Pseudomonas*, yeasts) with a high suppressive action (Table 4).

**Table 4.** Results of the microbiological and chemical analysis of the compost tea produced.

| Items | Value | Comment |
|---|---|---|
| *Escherichia coli* | Absent | |
| pH | 8.21 | |
| EC ($\mu$s cm$^{-1}$) | 2645 | Need of dilution before the use |
| Yeasts (c.f.u. mL$^{-1}$) | $2.53 \times 10^7$ | Action in biocontrol |
| Molds (c.f.u. mL$^{-1}$) | $4.00 \times 10^5$ | Filamentous fungi |
| *Bacillus* (c.f.u. mL$^{-1}$) | $4.33 \times 10^5$ | Antagonistic bacteria. Action in biocontrol |
| *Pseudomonas* (c.f.u. mL$^{-1}$) | $9.67 \times 10^4$ | Antagonistic bacteria. Action in biocontrol |
| Total bacteria (c.f.u. mL$^{-1}$) | $3.66 \times 10^5$ | |

The bio-stimulation tests on rocket seedlings also gave encouraging results. The compost extract, as it is, diluted 1:10 resulted in a 30% increase in the root system compared to the control (Table 5).

**Table 5.** Stimulating action of compost tea (CT) at different dilutions on the growth of rocket seedlings. (Ctrl: Control).

| Dilution | Root Length (R, mm) | Aerial Part Length (A, mm) | A/R |
|---|---|---|---|
| Ctrl | 25.37 | 24.07 | 0.95 |
| CT as it is | 16.88 | 32.05 * | 1.90 |
| 1:10 | 22.06 | 30.33 * | 1.37 |

**Table 5.** *Cont.*

| Dilution | Root Length (R, mm) | Aerial Part Length (A, mm) | A/R |
|---|---|---|---|
| 1:100 | 24.04 | 23.33 | 0.97 |
| 1:1000 | 20.25 | 20.29 | 1.00 |
| 1:10,000 | 22.07 | 18.17 | 0.82 |
| 1:100,000 | 21.18 | 17.26 | 0.81 |

* Statistically significant differences compared to the control-Dunnett test ($p < 0.01$).

### 3.2. Environmental Impacts

Environmental impacts per hectare varied among the analyzed systems depending on the impact category considered (Table 6). Indeed, SO2 was the most sustainable system with respect to metal depletion, climate change, photochemical oxidant, and particulate matter formation. In addition, in this system the use of corrugated boxes and the recycling end-of-life treatment of the materials provided carbon credits equal to –8% of the total impact. The most impacting systems, for almost all the impact categories considered, were those falling in the Basilicata region (RSC + CT; RSC; NRSC + CT; NRSC). In these regions, higher impacts for both systems grown on not replanted rows were found especially concerning to abiotic metal depletion, fossil depletion, and climate change. Therefore, the cultivation of rows already used in the previous production cycle seems to be more sustainable (Table 6).

**Table 6.** Environmental impacts per hectare divided by impact category and system analyzed (SC: conventional system; SI: integrated system; SO1: organic system; SO2: organic system; RSC + CT: replanted strawberry crop treated with compost tea; RSC: replanted strawberry crop without compost tea application; NRSC + CT: not replanted strawberry crop treated with compost tea; NRSC: not replanted strawberry crop without compost tea application).

| Impact Category | Unit | SC | SI | SO1 | SO2 | RSC + CT | RSC | NRSC + CT | NRSC |
|---|---|---|---|---|---|---|---|---|---|
| Climate change | kg $CO_2$ eq | 7461 | 9617 | 9116 | −972 | 11,068 | 10,996 | 11,431 | 11,359 |
| Ozone depletion | kg CFC-11 eq | 0 | 0 | 0 | 0 | 0 | 0 | 0 | 0 |
| Terrestrial acidification | kg $SO_2$ eq | 60 | 138 | 49 | 26 | 63 | 63 | 64 | 64 |
| Freshwater eutrophication | kg P eq | 3 | 0 | 1 | 7 | 6 | 6 | 6 | 6 |
| Marine eutrophication | kg N eq | 4 | 5 | 2 | 5 | 6 | 6 | 4 | 4 |
| Human toxicity | kg 1,4-DB eq | 7949 | 1118 | 7389 | 9983 | 14,641 | 14,633 | 14,545 | 14,537 |
| Photochemical oxidant formation | kg NMVOC | 33 | 17 | 26 | 7 | 49 | 47 | 52 | 49 |
| Particulate matter formation | kg PM10 eq | 11 | 15 | 10 | −10 | 18 | 18 | 18 | 18 |
| Terrestrial ecotoxicity | kg 1,4-DB eq | 380 | 3 | 3 | 37 | 393 | 392 | 389 | 389 |
| Freshwater ecotoxicity | kg 1,4-DB eq | 654 | 289 | 585 | 664 | 1261 | 1260 | 1232 | 1231 |
| Marine ecotoxicity | kg 1,4-DB eq | 6132 | 251 | 517 | 610 | 6546 | 6546 | 6540 | 6540 |
| Ionizing radiation | kBq U235 eq | 1328 | 771 | 968 | 1815 | 1701 | 1685 | 1689 | 1673 |
| Water depletion | m$^3$ | 1627 | 3125 | 3123 | 3334 | 9088 | 9085 | 9095 | 9092 |
| Metal depletion | kg Fe eq | −1045 | −3003 | 8313 | −6508 | −57 | −58 | −251 | −252 |
| Fossil depletion | kg oil eq | 4687 | 6128 | 5727 | 2087 | 5069 | 5064 | 5434 | 5429 |

In monetary units, the total environmental impact of strawberry cultivation was highly variable: it ranged from 1098 euros in SO2 to 6798 in RSC + CT and it was essentially influenced by the category with the most impact. Once again SO2 proved to be the most sustainable system (Table 7).

**Table 7.** Cost of the environmental impacts per hectare divided by impact category and system analyzed (SC: conventional system; SI: integrated system; SO1: organic system; SO2: organic system; RSC + CT: replanted strawberry crop treated with compost tea; RSC: replanted strawberry crop without compost tea application; NRSC + CT: not replanted strawberry crop treated with compost tea; NRSC: not replanted strawberry crop without compost tea application).

| Impact Category | Unit | SC | SI | SO1 | SO2 | RSC + CT | RSC | NRSC + CT | NRSC |
|---|---|---|---|---|---|---|---|---|---|
| Total | EUR2015 | 5508 | 2368 | 2086 | 1098 | 6798 | 6781 | 6780 | 6763 |
| Climate change | EUR2015 | 422 | 544 | 516 | −55 | 626 | 622 | 647 | 643 |
| Ozone depletion | EUR2015 | 0 | 0 | 0 | 0 | 0 | 0 | 0 | 0 |
| Terrestrial acidification | EUR2015 | 450 | 1033 | 365 | 194 | 473 | 470 | 482 | 478 |
| Freshwater eutrophication | EUR2015 | 6 | 1 | 3 | 13 | 11 | 11 | 10 | 10 |
| Marine eutrophication | EUR2015 | 11 | 16 | 7 | 16 | 19 | 19 | 13 | 13 |
| Human toxicity | EUR2015 | 711 | 100 | 661 | 892 | 1309 | 1308 | 1300 | 1300 |
| Photochemical oxidant formation | EUR2015 | 38 | 19 | 30 | 8 | 56 | 54 | 59 | 56 |
| Particulate matter formation | EUR2015 | 436 | 583 | 408 | −409 | 719 | 715 | 720 | 716 |
| Terrestrial ecotoxicity | EUR2015 | 3304 | 24 | 28 | 325 | 3412 | 3411 | 3378 | 3377 |
| Freshwater ecotoxicity | EUR2015 | 24 | 10 | 21 | 24 | 46 | 45 | 44 | 44 |
| Marine ecotoxicity | EUR2015 | 45 | 2 | 4 | 5 | 48 | 48 | 48 | 48 |
| Ionizing radiation | EUR2015 | 61 | 36 | 45 | 84 | 78 | 78 | 78 | 77 |

Figure 2 shows that the environmental cost in the SC, SO2, RSC + CT, RSC, NRSC + CT, and NRSC systems was due to more than 70% of Terrestrial ecotoxicity and Human toxicity; while in SI mainly to Terrestrial acidification and Particulate matter formation; and in SO1 to Climate change, Human toxicity, and Particulate matter formation (Figure 2).

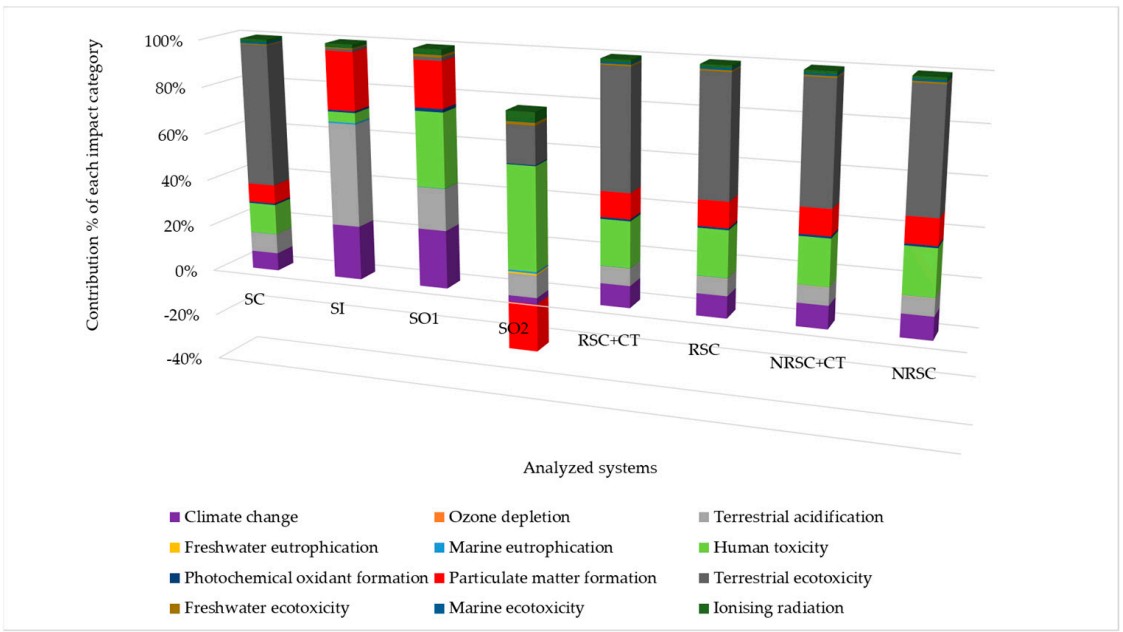

**Figure 2.** Contribution of each impact category to the cost of the total environmental impact for the different analyzed systems (SC: conventional system; SI: integrated system; SO1: organic system; SO2: organic system; RSC + CT: replanted strawberry crop treated with compost tea; RSC: replanted strawberry crop without compost tea application; NRSC + CT: not replanted strawberry crop treated with compost tea; NRSC: not replanted strawberry crop without compost tea application).



The disaggregation of the impacts of cultivation operations shows that, in almost all analyzed systems, soil disinfestation and materials used, namely iron structures and plastic (films, pipes, containers), were the items that caused the greatest environmental detriments (Figure 3). In systems where soil sanitation was not carried out with chloropicrin and dichloropropene (SI, SO1), irrigation and fertigation were the most impactful operations. The material used in SO2, a system in which the soil disinfestation was carried out with a natural method (solarization: a method of soil disinfestation based on solar heating by mulching soil with a transparent polyethylene during the hot season), represented more than 80% of the total impact. In particular, the zinc structures of the greenhouse and the use of corrugated boxes as containers for harvested strawberries were responsible for the greatest impacts. However, as already mentioned, the total recycling of these materials has allowed a significant reduction in the impacts.

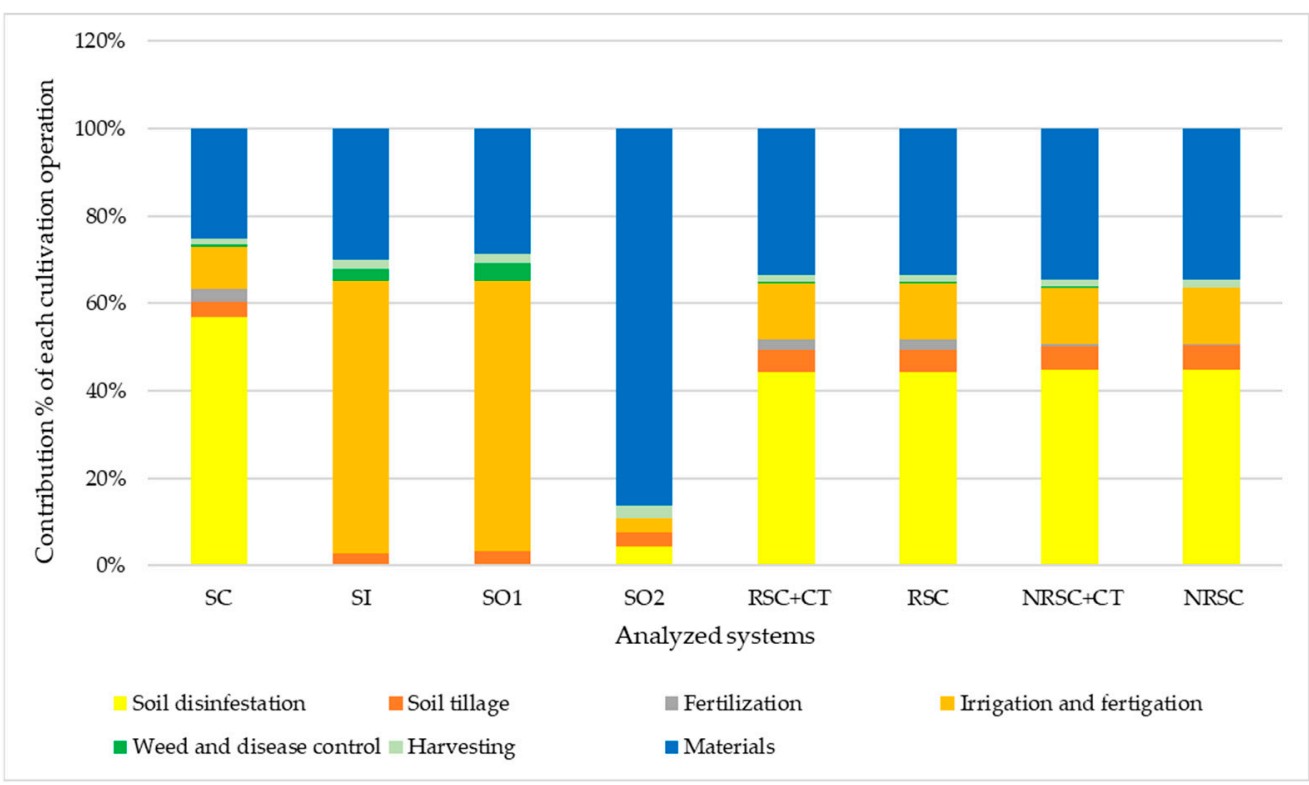

**Figure 3.** Contribution of each cultivation operation to the total environmental impact for the different analyzed systems (without considering the end-of-life recycling of the materials). (SC: conventional system; SI: integrated system; SO1: organic system; SO2: organic system; RSC + CT: replanted strawberry crop treated with compost tea; RSC: replanted strawberry crop without compost tea application; NRSC + CT: not replanted strawberry crop treated with compost tea; NRSC: not replanted strawberry crop without compost tea application).

CT production and its distribution on the field was by far the least impacting operation in the systems that used CT (RSC + CT, NRSC + CT), representing 1% of the impacts for all categories.

Data per kg of strawberries confirmed the sustainability of the SO2 system, while the lower productivity of SO1 made it the most impactful system. At the same time, in the two theses compared, the higher productivity due to the use of CT (Table 3) allowed for lower impacts per functional unit, referring to all impact categories (Table 8).

**Table 8.** Environmental impacts per kg of product divided by impact categories and strawberry systems analyzed (SC: conventional system; SI: integrated system; SO1: organic system; SO2: organic system; RSC + CT: replanted strawberry crop treated with compost tea; RSC: replanted strawberry crop without compost tea application; NRSC + CT: not replanted strawberry crop treated with compost tea; NRSC: not replanted strawberry crop without compost tea application).

| Impact Category | Unit | SC | SI | SO1 | SO2 | RSC + CT | RSC | NRSC + CT | NRSC |
|---|---|---|---|---|---|---|---|---|---|
| Climate change | kg $CO_2$ eq | 0.196 | 0.221 | 0.380 | −0.019 | 0.231 | 0.267 | 0.219 | 0.252 |
| Ozone depletion | kg CFC-11 eq | 0.000 | 0.000 | 0.000 | 0.000 | 0.000 | 0.000 | 0.000 | 0.000 |
| Terrestrial acidification | kg $SO_2$ eq | 0.002 | 0.003 | 0.002 | 0.000 | 0.001 | 0.002 | 0.001 | 0.001 |
| Freshwater eutrophication | kg P eq | 0.000 | 0.000 | 0.000 | 0.000 | 0.000 | 0.000 | 0.000 | 0.000 |
| Marine eutrophication | kg N eq | 0.000 | 0.000 | 0.000 | 0.000 | 0.000 | 0.000 | 0.000 | 0.000 |
| Human toxicity | kg 1,4-DB eq | 0.209 | 0.026 | 0.308 | 0.190 | 0.306 | 0.355 | 0.279 | 0.323 |
| Photochemical oxidant formation | kg NMVOC | 0.001 | 0.000 | 0.001 | 0.000 | 0.001 | 0.001 | 0.001 | 0.001 |
| Particulate matter formation | kg PM10 eq | 0.000 | 0.000 | 0.000 | 0.000 | 0.000 | 0.000 | 0.000 | 0.000 |
| Terrestrial ecotoxicity | kg 1,4-DB eq | 0.010 | 0.000 | 0.000 | 0.001 | 0.008 | 0.010 | 0.007 | 0.009 |
| Freshwater ecotoxicity | kg 1,4-DB eq | 0.017 | 0.007 | 0.024 | 0.013 | 0.026 | 0.031 | 0.024 | 0.027 |
| Marine ecotoxicity | kg 1,4-DB eq | 0.161 | 0.006 | 0.022 | 0.012 | 0.137 | 0.159 | 0.125 | 0.145 |
| Ionizing radiation | kBq U235 eq | 0.035 | 0.018 | 0.040 | 0.035 | 0.036 | 0.041 | 0.032 | 0.037 |
| Water depletion | $m^3$ | 0.043 | 0.072 | 0.130 | 0.064 | 0.190 | 0.220 | 0.174 | 0.202 |
| Metal depletion | kg Fe eq | −0.027 | −0.069 | 0.346 | −0.124 | −0.001 | −0.001 | −0.005 | −0.006 |
| Fossil depletion | kg oil eq | 0.123 | 0.141 | 0.239 | 0.040 | 0.106 | 0.123 | 0.104 | 0.121 |

In monetary units, the environmental cost per kg of strawberries harvested ranged from 0.02 euro (SO2) to 0.16 euro (RSC) (Table 9).

**Table 9.** Environmental costs per kg of product divided by impact categories and strawberry systems analyzed (SC: conventional system; SI: integrated system; SO1: organic system; SO2: organic system; RSC + CT: replanted strawberry crop treated with compost tea; RSC: replanted strawberry crop without compost tea application; NRSC + CT: not replanted strawberry crop treated with compost tea; NRSC: not replanted strawberry crop without compost tea application).

| Impact Category | Unit | SC | SI | SO1 | SO2 | RSC + CT | RSC | NRSC + CT | NRSC |
|---|---|---|---|---|---|---|---|---|---|
| Total | EUR2015 | 0.145 | 0.054 | 0.087 | 0.021 | 0.142 | 0.164 | 0.130 | 0.150 |
| Climate change | EUR2015 | 0.011 | 0.012 | 0.021 | −0.001 | 0.013 | 0.015 | 0.012 | 0.014 |
| Ozone depletion | EUR2015 | 0.000 | 0.000 | 0.000 | 0.000 | 0.000 | 0.000 | 0.000 | 0.000 |
| Terrestrial acidification | EUR2015 | 0.012 | 0.024 | 0.015 | 0.004 | 0.010 | 0.011 | 0.009 | 0.011 |
| Freshwater eutrophication | EUR2015 | 0.000 | 0.000 | 0.000 | 0.000 | 0.000 | 0.000 | 0.000 | 0.000 |
| Marine eutrophication | EUR2015 | 0.000 | 0.000 | 0.000 | 0.000 | 0.000 | 0.000 | 0.000 | 0.000 |
| Human toxicity | EUR2015 | 0.019 | 0.002 | 0.028 | 0.017 | 0.027 | 0.032 | 0.025 | 0.029 |
| Photochemical oxidant formation | EUR2015 | 0.001 | 0.000 | 0.001 | 0.000 | 0.001 | 0.001 | 0.001 | 0.001 |
| Particulate matter formation | EUR2015 | 0.011 | 0.013 | 0.017 | −0.008 | 0.015 | 0.017 | 0.014 | 0.016 |
| Terrestrial ecotoxicity | EUR2015 | 0.087 | 0.001 | 0.001 | 0.006 | 0.071 | 0.083 | 0.065 | 0.075 |
| Freshwater ecotoxicity | EUR2015 | 0.001 | 0.000 | 0.001 | 0.000 | 0.001 | 0.001 | 0.001 | 0.001 |

**Table 9.** *Cont.*

| Impact Category | Unit | SC | SI | SO1 | SO2 | RSC + CT | RSC | NRSC + CT | NRSC |
|---|---|---|---|---|---|---|---|---|---|
| Marine ecotoxicity | EUR2015 | 0.001 | 0.000 | 0.000 | 0.000 | 0.001 | 0.001 | 0.001 | 0.001 |
| Ionizing radiation | EUR2015 | 0.002 | 0.001 | 0.002 | 0.002 | 0.002 | 0.002 | 0.001 | 0.002 |

*3.3. Economic Results*

The most profitable systems were RSC + CT, RSC, NRSC + CT, and NRSC, namely those with the major total output and the lowest variable costs (Table 10), in which the gross profit per kg of strawberries was equal on average to 2.57 euros. SO1 was the less profitable system and even with a negative gross profit due to the low production and the consequently low TO which did not allow to cover the costs, which were the highest of all the systems.

**Table 10.** Economic results of the different analyzed strawberry systems. Values per year and hectare (SC: conventional system; SI: integrated system; SO1: organic system; SO2: organic system; RSC + CT: replanted strawberry crop treated with compost tea; RSC: replanted strawberry crop without compost tea application; NRSC + CT: not replanted strawberry crop treated with compost tea; NRSC: not replanted strawberry crop without compost tea application).

| | SC | SI | SO1 | SO2 | RSC + CT | RSC | NRSC + CT | NRSC |
|---|---|---|---|---|---|---|---|---|
| Average yield (kg) | 38,000 | 43,550 | 24,000 | 52,500 | 47,850 | 41,250 | 52,200 | 45,000 |
| Average price (€) [2] | 2.49 | 2.74 | 2.74 | 2.63 | 3.76 | 3.76 | 3.76 | 3.76 |
| Total Output (€) | 94,620 | 119,327 | 65,760 | 138,075 | 179,916 | 155,100 | 196,272 | 169,200 |
| Variable production costs (€) | 70,130 | 85,403 | 113,345 | 94,949 | 53,214 | 53,064 | 57,381 | 57,231 |
| Gross profit (€) | 24,490 | 33,924 | −47,585 | 43,126 | 126,702 | 102,036 | 138,891 | 111,969 |

Referring to the comparison of replanted and not replanted strawberry crop systems, producing on a new substrate ensured greater productivity and consequently a higher total output of about 16% more. At the same time, the use of CT allowed both systems to have an increase in production of 9% more and a related increase in TO. These systems recorded the lowest variable costs, equal on average to 55,000 €, slightly higher in NRSC systems (+8%), while almost equal with or without the use of CT (Table 10).

The breakdown of variable costs between the different agricultural operations showed that harvesting was the most expensive operation in SC, SI, and SO2, accounting for more than 60% of total costs. In the other systems, 80% of total costs were due, in addition to harvesting, plantation irrigation, and fertigation, above all in SO1 (Table 11).

**Table 11.** Breakdown of variable costs between the different agricultural operations. Values are per year and per hectare (SC: conventional system; SI: integrated system; SO1: organic system; SO2: organic system; RSC + CT: replanted strawberry crop treated with compost tea; RSC: replanted strawberry crop without compost tea application; NRSC + CT: not replanted strawberry crop treated with compost tea; NRSC: not replanted strawberry crop without compost tea application).

| | SC | SI | SO1 | SO2 | RSC + CT | RSC | NRSC + CT | NRSC |
|---|---|---|---|---|---|---|---|---|
| Soil preparation (€) | 3936 | 97 | 97 | 2249 | 8077 | 8077 | 8153 | 8153 |
| Plantation, irrigation system, and supporting structures installation (€) | 19,257 | 23,418 | 23,418 | 11,863 | 12,997 | 12,997 | 15,208 | 15,208 |
| Irrigation and Fertigation (€) | 951 | 5166 | 44,625 | 1189 | 13,391 | 13,391 | 13,391 | 13,391 |
| Weed and diseases control (€) | 346 | 1030 | 2739 | 12,000 | 651 | 651 | 651 | 651 |
| Compost tea production and distribution (€) | 0 | 0 | 0 | 0 | 150 | 0 | 150 | 0 |
| Harvesting (€) | 44,377 | 54,391 | 41,466 | 64,507 | 15,948 | 15,948 | 18,028 | 18,028 |
| Explant and other manual operations (€) | 1262 | 1300 | 1000 | 3142 | 2000 | 2000 | 1800 | 1800 |

The split of variable costs by single cost item showed that the purchase of seedlings was the most expensive cost item in almost all systems except in SO2, accounting on average for 20% of variable costs, followed by contracting services required for fumigation and mulching with plastic. Furthermore, other important costs that farmers had to bear were the cost of water for irrigation, of the containers (punnets) and boxes used to arrange the strawberries during harvesting, important above all in SO2, which accounted for more than 60% of the total cost.

SO1 proved to be the most uneconomical system per kg of produced strawberries, with a negative gross profit due to the low productivity but above all the higher production costs per functional unit (4.81 € kg$^{-1}$). If the environmental costs were added to the economic ones, the situation would become even more critical (Table 12). On the contrary, all the other systems—especially those in Basilicata and, among these, those that used CT—proved to be profitable and capable of covering the costs of pollution (Table 12).

**Table 12.** Economic results of the different analyzed strawberry systems. Values in € kg$^{-1}$. (SC: conventional system; SI: integrated system; SO1: organic system; SO2: organic system; RSC + CT: replanted strawberry crop treated with compost tea; RSC: replanted strawberry crop without compost tea application; NRSC + CT: not replanted strawberry crop treated with compost tea; NRSC: not replanted strawberry crop without compost tea application).

| System | Variable Costs | Environmental Cost | Total Cost | Total Output | Gross Profit |
|---|---|---|---|---|---|
| SO1 | 4.72 | 0.09 | 4.81 | 2.74 | −2.07 |
| SC | 1.85 | 0.14 | 1.99 | 2.49 | 0.50 |
| SI | 1.96 | 0.05 | 2.02 | 2.74 | 0.72 |
| SO2 | 1.81 | 0.02 | 1.83 | 2.63 | 0.80 |
| RSC | 1.29 | 0.16 | 1.45 | 3.76 | 2.31 |
| NRSC | 1.27 | 0.15 | 1.42 | 3.76 | 2.34 |
| RSC + CT | 1.11 | 0.14 | 1.25 | 3.76 | 2.51 |
| NRSC + CT | 1.10 | 0.13 | 1.23 | 3.76 | 2.53 |

## 4. Discussion

The present research aimed to evaluate the environmental sustainability and economic profitability of strawberry cultivation in the Campania and Basilicata regions (Southern Italy), whilst also considering the social cost of pollution. At the same time, the production effects of the application of CT on strawberry plants were tested. Particularly, two fields (one characterized by the presence of replanted rows and another by not replanted rows) were taken into account. The CT, used in this research, was obtained on a farm and it had a low production cost.

The environmental analysis highlighted the greater sustainability of the SO2 system which was characterized by high productivity, low pollution costs, and negative value of climate change impacts corresponding to carbon credits generation. The latter depended on the net reduction in the impact caused by the typology of packaging material used (cardboard) and its post-use treatment (100% recycling) [25]. Other important peculiarities of this system, which made it the most sustainable ever, was the method used for soil disinfestation (natural without the use of pesticides) and pest control through a biological fight, a good practice that should be spread as much as possible.

Under our experimental conditions, the strawberry cultivation emissions ranged from −0.019 to 0.380 kg $CO_2$ eq kg$^{-1}$, in SO2 and SO1, respectively. This last value was essentially due to the low productivity of the SO1 system. Consequently, all the analyzed systems were more sustainable when compared to other studies. Indeed, Parajuli et al. [25] calculated USA total GHG emissions per kg of strawberries and found a value equal to 1.45 kg $CO_2$ eq. Ilari et al. [27] in Central Italy found for the two systems analyzed (soilless and mulched soil tunnel) a similar impact per kg of strawberries produced, namely 0.785 kg $CO_2$ eq for the first and 0.778 kg $CO_2$ eq for the second; Tabatabaie and Murthy [3] observed

that the global warming potential for strawberry production varied from 1.75 to 5.48 kg $CO_2$ eq $kg^{-1}$ in California, Florida, North Carolina, and Oregon, while Mordini et al. [57] affirmed that in several countries this value ranged from 0.27 to 3.99 kg $CO_2$ eq $kg^{-1}$.

In our opinion, these results prove that in LCA studies, the choice of FU is essential. Indeed, the analysis per 1 kg of product leads to results that are not always reliable, since in equal impacts per hectare, the less productive crop results are more impactful. Moreover, differences in production practices and yields can cause great differences in climate change estimations [3], as well as the different methods for the calculation of emissions, the system boundaries chosen, and the life span of cropping systems (whole production cycle vs. 1 year of cultivation). Thus, there is a need to harmonize the approaches used in studies applying the LCA methodology within the agricultural sector. At the same time, the great variability that characterized the agronomic technologies and practices used does not allow us to define the standards above which crop systems have to be considered impacting.

The breakdown of emissions by production factors showed that plastic materials and zinc structures were the most impacting items, contrary to what we found in our other research, in which fuel consumption was the major cause of the total $CO_2$ eq emissions [14]. At the same time, the available studies report wide variations in the impacts, but some of them claim that most of the GHG emissions are attributed to the plastic used [25,31,33,34]. Furthermore, Peano et al. [58], found that, in the field phase, the use of a hose for irrigation and the PVC material used for mulching, account for the most significant impacts, and PE punnets and PE plastic film used for packaging represented, together, over 30% of the GWP. At the same time, in an Iranian study [32], in an open-field strawberry system, the contribution was mainly due to N and P fertilizers (around 60% of the total on-farm GHG emissions), followed by fuel consumption of farm machinery (about 20%); while in a greenhouse system, the consumption of electricity was the dominant contributor, followed by N-based fertilizers, irrigation water and the consumption of natural gas. Furthermore, Gunady et al. [33] and Maraseni et al. [59] identified that agricultural machinery operation, and the related fossil fuel consumption, was the 'hotspot' which accounted for 58% of the total GHG emissions, followed by chemical-fertilizer use (23%).

The use of a CT characterized by an elevated number of beneficial microorganisms with a high suppressive action allowed us to obtain, first of all, a good increase of the yield in both analyzed fields, confirming what has already been reported in the literature by other authors [34,37,39]. The qualitative analysis also showed an increase in the number of fruits, particularly red fruits, and their size. These are quality indicators that can complement the most important ones for both producers and consumers, such as aroma and sweetness [60]. Another important point in favor of the use of CT was the increase in profits for both fields, thanks to the aforementioned increase in yield. In particular, the systems analyzed in Basilicata were found to be the most profitable, on the one hand, due to the higher selling prices of strawberries found on the respective markets, and on the other hand, due to the lower production costs. On the contrary, the SO1 system proved to be the most expensive, above all for the considerable fertigation interventions carried out and therefore for the cost of the different inputs used.

Referring to the production costs, in our experimental conditions, seedlings, contracting services required for fumigation and mulching operations, water for irrigation, and packaging were the most expensive cost items. On the contrary, in an Iranian study [31], the higher variable expenditure in greenhouse-based production systems was mainly due to costs of natural gas and electricity while chemical fertilizers and machinery applied in open field cultivation were the two main variable costs. Moreover, Banaeian et al. [61], in Tehran province, found that among the cost inputs, transportation was the most important input followed by labor, fertilizers, and installation of equipment.

## 5. Conclusions

Under our experimental conditions, in which the assessments of environmental and economic aspects concerned only the cultivation of strawberries excluding the crop rotation

to improve soil fertility as in SI, SO1, and SO2, it emerged that organic crops are not always the most sustainable and profitable if significant irrigation and fertigation interventions are carried out. Apart from the management of the SO2 system, which is by far the most sustainable, the conventional system, characterized by few targeted interventions, resulted in the least impact both per hectare and per unit of product.

Moreover, the use of CT proved highly valid, and from a life cycle perspective, not impacting at all. The use of a quality green compost allowed us to obtain a good extract which increased the productivity of the systems analyzed, especially in the RSC system, and to have a higher gross profit. Furthermore, from an environmental point of view, the cultivation of rows already used in the previous production cycle was found to be more sustainable. At the same time, the validity of this technique lies in the finding of high-quality green compost, but at the same time, there is the need for evaluations on strawberry cultivars with different production times and with different phytopathological sensitivities. Furthermore, the sustainability assessment should be completed by extending the analyses to the crop rotations that characterize some systems.

**Author Contributions:** Conceptualization, M.P. and G.C. (Giuseppe Celano); methodology, M.P., M.Z., G.C. (Giuseppe Carlucci) and G.C. (Giuseppe Celano); software, A.M.; validation, M.P., and A.P.; formal analysis, G.C. (Giuseppe Carlucci) and M.Z.; investigation, G.C. (Giuseppe Carlucci), A.P. and G.A.; resources, G.C. (Giuseppe Celano); data curation, M.P. and A.M.; writing—original draft preparation, M.P.; writing—review and editing, A.M.P. and G.C. (Giuseppe Celano); visualization, M.P., G.C. (Giuseppe Celano), M.Z., A.M. and A.P.; supervision, G.C. (Giuseppe Celano); project administration, G.C. (Giuseppe Celano); funding acquisition, G.C. (Giuseppe Celano). All authors have read and agreed to the published version of the manuscript.

**Funding:** This research was funded by PSR Basilicata 2014–2020 Misura 16.2.1, "FOREST COMP Project—Valorizzazione degli scarti delle utilizzazioni forestali nella filiera compost on-farm" (Unique Project Code: H98D19001420006) and PON RESO—Sustainable and resilient horticulture and cereal food chain for the valorization of territories (Unique Project Code: B44I20000360005).

**Institutional Review Board Statement:** Not applicable.

**Data Availability Statement:** Not applicable.

**Acknowledgments:** We are grateful to Cooperative Sole—Parete (CE), MontellaBio srl—Frignano (CE), Azienda Agricola "Donatenato Sabato"—Scanzano Jonico (MT), Sele Natura Società Cooperativa Agricola—Eboli (SA) for giving us access to their field data.

**Conflicts of Interest:** The authors declare no conflict of interest.

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
