# Peer review of "An Environmental and Economic Analysis of Strawberry Production in Southern Italy"

_agriculture, doi:10.3390/agriculture13091705_

Round 1
Reviewer 1 Report
Dear authors,
Thanks a lot for your work. Indeed, significance of the topic addressed in this research is multifaceted. Firstly, the evaluation of the environmental and economic sustainability of strawberry cultivation holds paramount importance within the prevailing concerns regarding the environmental repercussions of agricultural practices and the pressing need for sustainable food production. Gaining insights into the sustainability of diverse cultivation systems is indispensable for informed decision-making and the identification of practices that mitigate adverse environmental impacts while ensuring economic viability.
Secondly, the investigation into the application of compost tea (CT) on strawberry plants assumes a crucial role as it explores a potential sustainable solution for augmenting productivity. The assessment of CT’s impact on strawberry yield offers invaluable insights into its viability as a sustainable agronomic practice.
This research makes a significant contribution by combining the evaluation of environmental and economic sustainability of strawberry cultivation with the investigation of compost tea application. While individual studies have been conducted on these topics, the integration of both aspects in a single study provides a comprehensive assessment that is relatively novel.
Overall, the relevance of the topic and the integration of sustainability evaluation and compost tea application make this research both timely and original, offering valuable insights for sustainable strawberry cultivation. Furthermore, considering the social cost of pollution in calculating the gross profit (GP) of farmers adds a socio-economic dimension to the evaluation of profitability. This aspect highlights the importance of incorporating externalities associated with pollution into economic analyses, offering a more comprehensive assessment of different cultivation systems' true costs and benefits.
Paper’s methodology is robust and clear. In particular, the inclusion of multiple systems allows for a comparative analysis, which is beneficial for assessing the sustainability and productivity of different approaches. Authors provided specific details regarding the selection and characterization of these systems.
The production and characterization of compost tea (CT) are described sufficiently.
The quantitative analysis of strawberry yield and the assessment of environmental impact using the life cycle assessment (LCA) methodology demonstrate a systematic approach to data collection and analysis.
The calculation of gross profit (GP) and the consideration of the social cost of pollution in profitability calculations indicate that the economic analysis included both direct and indirect costs. This approach provides a more comprehensive understanding of the economic viability and sustainability of the different strawberry growing systems.
However, there are some suggestions to improve the paper’s quality:
1) It would be better to add some estimates on consumers and producers awareness of the environmental impact of stuff productive processes in INTRODUCTION.
2) Table 5: data overlaps with rows numbers.
1) Table 1: Please clarify “-” sign for Soil management/ weed control data for SC, SI, SO1 (no data or no application).
2) Think about presentation of text in 162-199 rows in a graph. This text contains lots of technical details making it difficult to read and understand.
3) Authors mentioned that the applied method excluded assessment of climate change impact (256-258), but there are estimates for climate change impacts in the results. So, please, clarify the climate change impact assessment method.
Author Response
Please, see the attachment.

Reviewer 2 Report
This is an interesting manuscript that describes a study that investigated the environmental and economic aspects of strawberry production following applications of compost tea. It is very hard to read due to poor use of the English language – mainly sentence construction and grammar. There are far too many errors to list individually. There are also several non-specific and unscientific phrases such as ‘stuff’ at Line 46 – replace ‘stuff’ with ‘commodity. The manuscript needs to be proof-read and corrected by a native English speaker or sent to a professional editing service.
There are abbreviations in the Abstract which are not necessary. Readability would be improved if these are removed. Indeed, abbreviations should be kept to a strict minimum.
The term ‘sustainability’ in modern science implies a balance between economic growth, environmental responsibility and social well-being. Economics and environmental issues are not independent and should not be considered so in a sustainability study. Therefore, the opening statement of the abstract (lines 16-19) needs to be rewritten e.g. ‘This article aims to provide an evaluation of the environmental and economic aspects of strawberry cultivation in Campania and Basilicata regions of Southern Italy, and 2) to consider the effects on strawberry productivity following the application of composted tea.’
Secondly, this is not really new science as there have been many similar studies over the years that focus on the sustainability of strawberry production. Some are discussed briefly in the introduction but there are many, many more. The authors need to emphasise what new information this article adds to the science base. This is particularly important as the authors use the term ‘innovative’ in the title but the innovation is far from obvious.
At line 102 the authors state they are not aware of any previous study that considers the environmental and economic aspects of strawberry production. The authors are misinformed. A quick look on Google scholar identifies many such studies (e.g. Banaeian et al., 2011; Rysin et al., 2015; Mousavi et al., 2023; Gendron et al., 2017; Jadhav et al., 2017; Claire et al., 2018; etc.). The authors have cited a couple of these but not in the context of this statement. This statement needs to be reworded and a much stronger rationale for the study needs to provided.
A map of the area would be useful. Be careful not to duplicate information in the Tables and main text e.g. Table 1.
At Line 142 the authors mention data collection much more detail of this is needed. How were farmers identified and how many? What type of direct interviews? What data was collected how was this analysed? Were there data gaps and how were these treated? How was bias avoided?
How was the compost tea sampled and analysed?
The limitations of the study including those embedded in the LCA approach need to be discussed in full, especially the use of the data within the Simapro databases which have significant limitations.
I have concerns regarding the economic analysis. Prices, taxes, duty etc. are extremely volatile and vary considerably from one region/country to another. This part of the study is a narrow snapshot which is already well out of date, which makes many of the conclusions of the study irrelevant especially as this is an international journal. Whilst this is touched upon at the end of Section 4 the discussion is far from adequate. This needs to be discussed in detail.
See section above.
Author Response
Please, see the attachment.

Reviewer 3 Report
This research is a topic of great practical significance, using life cycle assessment of the impact of compost tea application on strawberry production, including environmental impact and economic sustainability analysis. The discussion section is well analyzed. The conclusion section is based on the results, from specific to general summary. Here are some suggestions for the author's reference.
(1) The abstract part is too long-winded. The innovation and practical significance of the article should be summarized more, and the relevant conclusions can be deleted to highlight the main conclusions.
(2) The introduction part lacks the clarification and summary of the existing research, and the scientific issues and innovation points are not clearly explained.
(3) The part of materials and methods can be appropriately narrowed, and it is enough to clearly explain the data source and processing process.
(4) There are format problems in some tables, such as Table 5, which need further proofreading.
Author Response
Please, see the attachment.
